# Advantage of Dimethyl Sulfoxide in the Fabrication of Binder-Free Layered Double Hydroxides Electrodes: Impacts of Physical Parameters on the Crystalline Domain and Electrochemical Performance

**DOI:** 10.3390/ijms231710192

**Published:** 2022-09-05

**Authors:** Gayi Nyongombe, Guy L. Kabongo, Luyanda L. Noto, Mokhotjwa S. Dhlamini

**Affiliations:** Department of Physics, College of Science Engineering and Technology, University of South Africa, Private Bag X6, Florida, Science Campus, Christiaan de Wet and Pioneer Avenue, Florida Park, Johannesburg 1710, South Africa

**Keywords:** dimethyl sulfoxide, layered double hydroxides, binder-free LDH electrode, supercapacitor

## Abstract

The electrode fabrication stage is a crucial step in the design of supercapacitors. The latter involves the binder generally for adhesive purposes. The binder is electrochemically dormant and has weak interactions, leading to isolating the active material and conductive additive and then compromising the electrochemical performance. Designing binder-free electrodes is a practical way to improve the electrochemical performance of supercapacitors. However, most of the methods developed for the fabrication of binder-free LDH electrodes do not accommodate LDH materials prepared via the co-precipitation or ions exchange routes. Herein, we developed a novel method to fabricate binder-free LDH electrodes which accommodates LDH materials from other synthesis routes. The induced impacts of various physical parameters such as the temperature and time applied during the fabrication process on the crystalline domain and electrochemical performances of all the binder-free LDH electrodes were studied. The electrochemical analysis showed that the electrode prepared at 200 °C-1 h exhibited the best electrochemical performance compared to its counterparts. A specific capacitance of 3050.95 Fg^−1^ at 10 mVs^−1^ was achieved by it, while its Rct value was 0.68 Ω. Moreover, it retained 97% of capacitance after 5000 cycles at 120 mVs^−1^. The XRD and FTIR studies demonstrated that its excellent electrochemical performance was due to its crystalline domain which had held an important amount of water than other electrodes. The as-developed method proved to be reliable and advantageous due to its simplicity and cost-effectiveness.

## 1. Introduction

The electrode fabrication stage is one of the major steps in the development of technological devices, such as supercapacitors and batteries [1,2]. It generally includes three main steps, such as (1) the mixture of the active material, conductive additive and binder with consideration to the ratio of each component into a solvent to make the slurry [3]; (2) the coating/casting/dropping of the slurry onto a selected substrate; and (3) the drying of the as-prepared electrode at a selected temperature and time [4,5]. Normally, binders are involved during the fabrication of electrodes for adhesive purposes. However, binders are generally dead mass and electrochemically inert. Consequently, they isolate the active materials as well as the conductive additives. As a result, they compromise the electrochemical performance of energy storage devices [3]. To solve this issue, the development of binder-free electrodes is highly encouraged [5,6,7,8,9]. The fabrication of binder-free electrodes is a practical way to improving the electrochemical performance of supercapacitors as well as batteries and reduce the production cost. Compared to traditional electrodes, binder-free electrodes possess distinctive advantages for electrochemical energy storage applications among which high mass loading of active materials, a good connection between the active materials and current collectors since the binder is absent, full utilization of active materials, and efficient diffusion of electrons and ions within the electrodes [3]. Currently, several methods to fabricate binder-free electrodes have been developed such as chemical vapor deposition [10], vacuum filtration [11], hydrothermal/solvothermal [12], aerogel production [13], electrospinning [14], electrochemical deposition, and electrophoretic [3,15]. The fabrication process of binder-free electrodes generally relies on the direct growth of the electrode active material onto a specific conductive substrate such as carbon cloth [16], carbon paper [17] and nickel foam [18]. However, compared to other conductive substrates, nickel foam has been extensively used as a substrate to make binder-free electrodes due to its advantages such as high electronic conductivity, high specific surface area, open-pore structure, good mass transport, and micro-holes. More importantly, it is cheap compared to other conductive substrates [18].

Nickel foam is generally manufactured through the coating of Ni metal on a polymer substrate via chemical vapor deposition (CVD) or electrochemical deposition techniques [19,20]. Normally, the pore size of a nickel foam depends on the polymer’s arrangement even though it is commonly ranged from 5 to 130 pores per inch [18]. More interestingly, the nickel foam’s architecture which is usually 3D favors a high specific surface area on which the electrochemical reactions take place. This makes nickel foam to be a desirable conductive substrate for many applications. The hydrothermal or solvothermal techniques were reported to be facile and promising routes for the direct growth of the active material on the nickel foam. As a result, nickel foam has been the most extensively used conductive substrate compared to others [18].

Layered double hydroxides (LDH) are lamellar inorganic solids considered as promising electrode active materials for supercapacitors due to their excellent electrochemical property which is the result of combined impacts of two or more metal cations involved during the synthesis [21]. Adding to this, various advantages such as facile synthesis, unique structure, unvarying distribution of diverse metal cations in the brucite layer, surface hydroxyl groups, high tunability, intercalated are anions with interlamellar spaces, excellent chemical stability, and the ability to intercalate diverse varieties of anions (inorganic, organic, biomolecules, and even genes) make them a center of great research attraction [22]. Several LDH materials have recently been directly synthesized on nickel foam and applied as binder-free electrodes for supercapacitor applications. However, the majority of them were hydrothermally fabricated [4,7,8,23,24,25]. This shows how the current fabrication process of nickel foam-based binder-free LDH electrodes do not accommodate LDH materials synthesized through other methods such as the co-precipitation and ion exchange; whereas the latter are the most used techniques to prepare LDH materials [26,27].

We recently reported on the benefits of dimethyl sulfoxide (DMSO) used as a binder solvent on the electrochemical performance of layered double hydroxides (LDH) [28]. During the experiment, it was noticed that DMSO prevents the LDH electrode crystalline structure from great damage due to the electrode drying parameters. More importantly, the adhesive property of the slurry prepared using DMSO on the nickel foam was remarkably high which could probably be due to both DMSO and the electrode drying temperature. Inspired by this observation and considering that W. Blake Hawley et al., had demonstrated that the casting temperature can ameliorate the adhesive property of a slurry, favoring a speed coating and a successful vacuum pressure [29]. We intended to fabricate LDH electrodes without involving the binder while DMSO was used as a binder solvent. Benefiting from the spectacular adhesive property of DMSO due to the electrode drying temperature, several binder-free LDH electrodes deposited on nickel foam were fabricated. Afterwards, the impacts of physical parameters such as the electrode drying temperature and time on the crystalline domain and electrochemical performances of the as-fabricated electrodes were investigated. More interestingly, we developed a novel cost-effective method for the growth of LDH materials on nickel foam without binders. More importantly, this technique can accommodate LDH materials prepared via other synthesis methods such as the co-precipitation, and exchange ions.

## 2. Results and Discussion

The crystalline structure is an important part for LDH materials. It was reported that during the charging step, the surface of LDH materials is not the only part that participates, the entire crystalline structure is also involved via intercalation/de-intercalation of electrolyte ions; promoting the excellent energy storage capabilities of LDH materials [6,30,31,32,33]. Therefore, the LDH crystalline domain deserves attention. Figure 1 shows the XRD patterns of the as-synthesized LDH used as the electrode active material. Even though this result was discussed in detail in our previous work [28], however, all the diffraction peaks depicted could be attributed to the planes of layered hydrotalcite-like material [5,34].

Subsequently, Figure 2a,b display the schematic illustration of the fabrication process of binder-free LDH electrodes as well as the image of the as-fabricated binder-free LDH electrode. Figure 2c shows the XRD patterns of the LDH-100-1h, LDH-100-1h30, LDH-100-2h, LDH-150-1h, and LDH-200-1h electrodes. Compared to the XRD patterns of the as-synthesized LDH, the XRD patterns of all the binder-free LDH electrodes displayed types of disordered stacked structures which could be due to the effects of electrode drying parameters [28,35]. Nevertheless, some diffraction peaks corresponding to the plane of layered hydrotalcite labelled with red starts were observed in the XRD patterns of all the binder-free LDH indicating that the initial crystalline structures were of the planes of layered hydrotalcite-like material, thereafter, they got altered probably due to the applied physical parameters. The LDH diffraction peaks recorded from the XRD patterns of all the binder-free LDH electrodes were positioned at 24°, 34°, and 39° indexed to (006), (012), and (015) [5,34]. Adding to this, a diffraction peak located at 11° indexed to (003) was noticed, but only for the LDH-150-1h electrode. Furthermore, diffraction peaks depicted at 60° indexed to (110) were also visible for the LDH-100-1h, LDH-100-1h30, and LDH-150-1h electrodes [5,34]. Apart from LDH diffraction peaks, two other diffraction peaks with high intensities were depicted for all the as-fabricated binder-free LDH electrodes and assigned to nickel foam [36,37,38]. In addition, the remaining diffraction peaks observed for all the electrodes beside those highlighted above were considered as foreign peaks resulting from the electrode fabrication process. More interestingly, it was reported that the effects of structural disorder can be very beneficial for an energy storage device since it can improve the electrochemical activity [39,40]. The difference noticed on the intensity or appearance of diffraction peaks could be attributed to the applied physical parameters. It is known that the LDH crystalline domain deserves an attention, unfortunately, in this work, the very low intensity of LDH diffraction peaks recorded for all the binder-free LDH electrodes made the deep analysis of XRD profiles difficult. As a result, the Fourier transform infrared (FTIR) measurements were performed for all the binder-free LDH electrodes in order to understand the impacts of applied physical parameters on the interlamellar for all the binder-free LDH electrodes.

Figure 2d shows the FTIR spectra for the LDH-100-1h, LDH-100-1h30, LDH-100-2h, LDH-150-1h, and LDH-200-1h electrodes. Generally, the nature of the LDH interlamellar also determines the electrochemical performance [30,41]. Since it was difficult to understand the nature of the interlamellar for all the as-fabricated binder-free LDH electrodes using XRD, the FTIR measurements for all the binder-free LDH electrodes were carried out. To understand the FTIR results, the focus was given to three major regions pointed out by arrows with different colors. The arrows with the color red refer to the vibration bands ranging from 3000 to 3600 cm^−1^ that were attributed to the O-H stretch of physically adsorbed water, water molecules within the interlamellar, and the hydroxyl group [41]. The blue colored arrows refer to the physically adsorbed water stretch; while the pink colored arrows refer to carbonate vibrations [42]. Enlargement of curves of vibration bands assigned to the O-H stretch of physically adsorbed water, water molecules within the interlamellar, and the hydroxyl group were noticed for the LDH-100-2h and LDH-200-1h electrodes compared to the LDH-100-1h, LDH-100-1h30, LDH-150-1h electrodes (refer to the red colored arrows). It must be noted that the width or intensity of an infrared band is the consequence of the chemical environments within the materials. This reflects also the strength of intermolecular interactions [43]. Nyongombe et al. have recently demonstrated that the electrode drying temperature is one of the major factors that triggers changes in the crystalline domain of LDH electrodes [28]. This leads to the state that due to the electrode drying temperatures, the hydrogen bonds in molecules of water were differently affected resulting in different strengths of interactions of intermolecular within binder-free LDH electrodes that caused the curves of vibration bands ranging from 3000 to 3600 cm^−1^ to differ in shape [41]. This can be confirmed by the difference in recorded intensities of bands attributed to vibration of physically adsorbed waters (refer to the blue colored arrows) [42]. After a deep analysis of bands attributed to the vibration of physically adsorbed waters, it was observed that the LDH-200-1h electrode possesses a band with high intensity compared to its counterparts, followed by the LDH-100-2h, LDH-100-1h, LDH-150-1h, LDH-100-1h30 electrodes, respectively. This reveals the difference in the amount of physically adsorbed water within the interlamellar of all the electrodes [42]. It was also recently proved that during the LDH electrode fabrication, the molecule from the solvent used can easily be intercalated in the interlamellar due to the impact of the electrode drying temperature and alter the crystalline domain [28]. Considering this and the reagents used during the fabrication of all the electrodes, it can be assumed that sulfur particles were released from DMSO due to the electrode drying temperatures according to Equation (1) [44] and got encapsulated in carbon matrix from the conductive additive, then both got intercalated into the interlamellar of all the electrodes [45,46]. Consequently, the interlamellar environment was greatly altered compared to a pure LDH, causing the diffraction peaks indexed to (003) to be invisible in the XRD patterns of all the electrodes. This hypothesis can also be supported by the behaviors of carbonate anions in the interlamellar of all the electrodes (refer to the pink colored arrows). It is known that more than one atomic species in the interlamellar of LDH possesses a carbonate anion $\left({\mathrm{CO}}_{3}^{2-}\right)$ and the corresponding bands are generally distinguished by a D3 h trigonal planar symmetry that shows the vibrations V2, V3, V4 which can be located at 860, 1360, 774 cm^−1^, respectively [42,47]. In this work, the recorded bands attributed to carbonate vibrations ranged from 1355 to 1365 cm^−1^ for all the electrodes indicating the interactions between carbonate groups and water molecules in the interlamellar. It also demonstrates the existence of fractions of carbonates in the lower symmetry [42,48]. A careful analysis of bands attributed to carbonate vibrations revealed a difference in their width and intensities. It was observed that the LDH-200-1h electrode possesses a wide and intense band compared to its counterparts, followed by the LDH-100-2h, LDH-150-1h, LDH-100-1h, LDH-100-1h30 electrodes, respectively. This reveals the interactions of different forms of carbonate anion. It further informs about the amount of low symmetry carbonate anions in the interlamellar for all the electrodes [42,47]. Therefore, it can be stated that the LDH-200-1h electrode possesses a high amount of low symmetry carbonate anions in the interlamellar compared to its counterparts. Followed by the LDH-100-2h, LDH-150-1h, LDH-100-1h, LDH-100-1h30 electrodes, respectively. Moreover, shoulders at 1427 and 1469 cm^−1^ were also depicted for the LDH-100-2h and LDH-100-1h30 electrodes, respectively. These could be attributed to the characteristics of free low symmetry carbonate anion on the surface [47].
(1)$$2\left({\mathrm{CH}}_{3}\right)2\mathrm{SO}\to \left({\mathrm{CH}}_{3}\right)2{\mathrm{SO}}_{2}+\left({\mathrm{CH}}_{3}\right)2S$$

Afterwards, the morphologies of all the as-fabricated binder-free LDH electrodes were captured as depicted in Figure 3a–f. The latter compared the morphology of the as-synthesized LDH used as electrode active material to those of the as-fabricated binder-free LDH electrodes. The flower-like structures made of nanosheets were noticed as morphology for all the binder-free LDH electrodes which were comparable to the morphology of the as-synthesized LDH used as electrode active material. However, it was observed that the leaves of flowers were opened for the morphologies of all the as-fabricated binder-free LDH electrodes compared to those of the as-synthesized LDH used as electrode active material. Subsequently, Figure 4a–e display the morphologies of all the as-fabricated binder-free LDH electrodes at a different magnification revealing the adhesion of slurries on nickel foam. The mass-loading for the as-fabricated binder-free LDH electrodes is displayed in Table 1. It was noticed that two sets of electrodes possess comparable masses. Understanding the casting step can allow the fabrication of electrodes with constant mass. These results indicate that the applied physical parameters had more effect on the structural domains than the morphologies.

The electrochemical analyses were performed to track the effects of physical parameters applied during the electrode fabrication on the electrochemical performances of the LDH-100-1h, LDH-100-1h30, LDH-100-2h, LDH-150-1h, and LDH-200-1h electrodes. The cyclic voltammetry (CV) measurements were carried out in 1.0 M KOH electrolyte in a three-electrode system. Figure 5a displays the comparative CV curves for the LDH-100-1h, LDH-100-1h30, and LDH-100-2h electrodes at the scan rate of 10 mVs^−1^ in a voltage window of 0.6 V (vs. Ag/AgCl). As it can be seen, typical Faradaic peaks were noticeable for all the electrodes showing that their capacitances were the consequences of quasi-reversible faradaic redox reactions [30,49,50] due to the combined effects of nickel and cobalt within the LDH electrodes [51]. Moreover, it was depicted that the CV absolute area of the LDH-100-2h electrode was larger compared to those of the LDH-100-1h and LDH-100-1h30 electrodes indicating that its charge storage performance is excellent compared to its counterparts. Subsequently, the CV curve of the LDH-100-2h electrode was compared to those of the LDH-150-1h and LDH-200-1h electrodes at 10 mVs^−1^ as shown in Figure 5b. It is obvious that the CV absolute area of the LDH-200-1h electrode is larger compared to those of the LDH-100-2h and LDH-150-1h electrodes indicating that the LDH-200-1h electrode possesses a high energy storage capacity compared to the LDH-100-2h and LDH-150-1h electrodes. This was followed by the LDH-100-1h and LDH-150-1h electrodes, respectively. Moreover, peaks related to the Faradaic redox reactions were also noticed in the CV curves of the LDH-150-1h and LDH-200-1h electrodes.

Afterwards, Figure 6a–e show the CV curves for all the electrodes at different scan rates (10, 30, 50, 75, and 100 mVs^−1^) in the voltage window of 0.6 V. No misshaping was observed for all the electrodes as the scan rate was increasing from 10 to 100 mVs^−1^, indicating a relatively high-current capability [22]. Studying electrode kinetics mechanisms is crucial due to the interfacial nature of electrochemistry. Therefore Equations (2) and (3) [28] were applied for this purpose.
(2)$$i=icap+idiff=av^{b\;}$$
(3)logi=loga+blogv

Consequently, Figure 7a–e display the dependence of anodic peak current (ipa) on the square root of the scan rate for all the electrodes revealing values of their R^2^ which were considered as their b values. The results demonstrate that the reaction mechanisms for all the electrodes were governed by the surface capacitance and diffusion-controlled processes. Thereafter, Equations (4) and (5) [28] were used to estimate the contribution of each process in the overall kinetic mechanism for all the electrodes.
(4)$$i=k_{1\;}V+k_{2}V^{1/2\;}$$
(5)iV1/2=k1 V1/2+k2

Using Equation (5), plots were drawn with *i/V*^1/2^ against *V*^1/2^ for all the electrodes as displayed in Figure 8a–e. Then after the linear fit of plots, the values of their slopes were taken as their *k*_1_ while the intercepts were their *k*_2_ [28]. Multiplying the values of their *k*_1_ and *k*_2_ by the scan rate of 10 mVs^−1^, the contribution fractions of the surface capacitance and diffusion-controlled processes and their percentage at the scan rate of 10 mVs^−1^ for all the electrodes were revealed in Figure 8f. From the results, it was noticed that the diffusion-controlled process contributed the most to the overall charge storage mechanisms for all the electrodes showing that the dominant mechanisms were battery-like [28]. However, the percentage of the contribution of the diffusion-controlled process to the overall charge storage mechanism was different from one electrode to the other, probably due to the nature of their interlamellar domain.

Thereafter, the specific capacitances for the LDH-100-1h, LDH-100-1h30, LDH-100-2h, LDH-150-1h, and LDH-200-1h electrodes were calculated from CV curves at the scan rate of 10 mVs^−1^ using Equation (6) [30], and the calculated values are presented in Table 2. Meanwhile, Table 3 compared the specific capacitance of the LDH-200-1h electrodes with other reported NiCoAl-LDH electrodes.
(6)$$Csp=\frac{1}{Vm∆Vif}\int_{Vi}^{Vf}\left(E\right)dE$$
where *Csp* is the specific capacitance (Fg^−1^), *V* is the scan rate (Vs^−1^), *m* is the mass of active material on the substrate (g), ∆*V* is the potential window applied for the measurements (*Vi* to *Vf*), and the integral term is the absolute area of the CV curve.

The electrochemical impedance spectroscopy (EIS) measurements were also performed for the LDH-100-1h, LDH-100-1h30, LDH-100-2h, LDH-150-1h, and LDH-200-1h electrodes in a frequency ranging from 100 kHz to 100 mHz at open circuit potential. It is known that Rs shows the resistance of the electrolyte, while the charge transfer impedance on the interface electrode/electrolyte is represented by Rct [30,52]. Figure 9a exposes the comparative Nyquist plots for the LDH-100-1h, LDH-100-1h30, LDH-100-2h electrodes. All the electrodes displayed slight semicircles in the high-frequency region, whereas the almost straight lines were observed in the low-frequency region. Generally, a slight semicircle in the high-frequency region indicates the faradaic reaction [53]. While the almost straight line in the low-frequency region informs about the diffusion of redox species and their kinetics [30,54]. The insert in Figure 9a shows the zoomed comparative Nyquist plots for the LDH-100-1h, LDH-100-1h30, LDH-100-2h electrodes. As it can be seen, the LDH-100-2h electrode exhibited a low Rct compared to its counterparts, followed by the LDH-100-1h electrode. Thereafter, the Nyquist plot of the LDH-100-2h electrode was compared to those of the LDH-150-1h and LDH-200-1h electrodes as displayed in Figure 9b. However, it is noticeable from the zoomed comparative Nyquist plots shown in the insert in Figure 9b that the LDH-200-1h electrode possesses a low Rct compared to the LDH-150-1h and LDH-100-2h electrodes; the recorded Rct values for all the electrodes are displayed in Table 4.

The electrochemical studies revealed that the LDH-200-1h electrode is the best compared to its counterparts. Followed by the LDH-100-2h, LDH-100-1h, LDH-100-1h30, and LDH-150-1h electrodes, respectively as shown in Figure 10a. This could be attributed to the impacts of the physical parameters used during their fabrication. It is obvious that the physical parameters applied for the LDH-200-1h electrode have favored its crystalline structure to hold an important amount of water compared to other electrodes (refer to Figure 2d). However, the hydrophilic nature of the crystalline structure of LDH improves the ionic diffusion and contributes greatly to the electrochemical performance [6,30,31,32,33]. It was recently demonstrated that when the LDH crystalline structure hydrophilicity is reduced, the basal spacing also decreases [41]. Adding to this, studies have proved that a larger basal spacing is very important because it favors the electrolyte ions penetration during the charging storage, which results in optimizing the electrochemical performance [28,30,41]. That is the reason that favored the excellent electrochemical performance of the LDH-200-1h electrode compared to other electrodes. Furthermore, Table 5 compares the EIS results recorded in this work to those of the NiCoAl-NMP and NiCoAl-DMSO electrodes recently reported [28], it is obvious that those recorded in this work exhibited very low Rct compared to the NiCoAl-NMP and NiCoAl-DMSO electrodes [28], except from the LDH-150-1h electrode. This could be attributed to high electrical conductivity that resulted from the effect of carbon black used as a conductive additive and the absence of the binder. Moreover, it could also be assumed that the physical parameters have favored a strong combination between sulfur and carbon within the LDH crystalline domain which had a good impact on the electrochemical performance of some electrodes [45]. However, the physical parameters applied for the LDH-150-1h electrode have greatly damaged the crystalline structure which had compromised its electrochemical performance. Furthermore, an electrode is generally acknowledged promising if it exhibits excellent specific capacitance and long-term cycling stability. Consequently, the electrochemical cycling stability of the LDH-200-1h electrode was studied by CV [59,60,61]. Figure 10b displays the plot of capacitance retention against the cycle number. As it can be seen, the LDH-200-1h electrode retained 97% of its capacitance after 5000 cycles. The insert in Figure 10b shows CV curves of the LDH-200-1h electrode at the scan rate of 120 mVs^−1^ from 1st to 5000th cycles in a 1 M KOH solution. No change of position was noticed in peaks potential indicating excellent electrochemical reversibility [60] which could be attributed to the nature of its crystalline domain as well as to a good intimate contact between the assumed sulfur particles and carbon within its interlamellar [62].

## 3. Materials and Methods

### 3.1. Binder-Free LDH Electrodes Preparation

Binder-free LDH electrodes were fabricated by mixing the LDH material and carbon black in a ratio of (80:20) into DMSO solvents. The LDH material used as active material in this study was taken from our recently reported work, whereby its synthesis procedure was explained in detail [28]. The slurry was obtained with the assistance of ultrasonication for 15 min. Thereafter, five cleaned 1 cm × 1 cm pieces of nickel foam were immersed inside the vial containing the slurry and aged for 48 h at room temperature. Afterwards, three of them were selected and dried at 100 °C while the time was varying from 1 h, 1 h 30 min to 2 h. The remaining two were then dried for 1h at different temperatures such as 150 °C and 200 °C. Subsequently, they were labelled according to the temperature and time applied during their fabrication and their mass-loading as displayed in Table 1. The binder-free LDH electrodes dried at 100 °C for different times were named as follows: LDH-100-1h, LDH-100-1h30, and LDH-100-2h respectively. Whereas dried for 1h at different temperatures were labelled as LDH-150-1h and LDH-200-1h respectively.

### 3.2. Materials Characterization

The Rigaku Smartlab diffractometer with (λ = 0.15405 nm) was used to collect XRD patterns of the as-synthesized LDH as well as all the as-fabricated binder-free LDH electrodes. While the FT-IR studies of the as-obtained LDH and all the as-fabricated binder-free LDH electrodes were performed using the IR Tracer-100-SHIMADZU (3750–500 cm^−1^). The morphologies of the as-prepared LDH and all the as-fabricated binder-free LDH electrodes were captured using a scanning electron microscope (SEM-EDS JEOL JSM-7800F) coupled with an EDS detector. In addition, the electrochemical data were collected on an Autolab PGSTAT302N potentiostat using a three-electrode system. All the binder-free LDH electrodes, platinum wire, and Ag/AgCl (3 M KCl-filled) were used as working, counter, and reference electrodes, respectively. 1 M KOH solution was used as an electrolyte. Finally, the electrochemical impedance spectroscopy (EIS) measurements were conducted with an AC amplitude of 5 mV in the frequency range of 100 kHz–100 mHz.

## 4. Conclusions

This work revealed the benefits of DMSO in the fabrication of binder-free LDH electrodes and it also demonstrated a novel technique to fabricate binder-free LDH electrodes which accommodates LDH materials prepared via other synthesis routes such as the co-precipitation and ions-exchange methods. A series of physical parameters was applied during the fabrication of binder-free LDH electrodes such as 100 °C-1h, 100 °C-1h30, 100 °C-2h, 150 °C-1h, and 200 °C-1h. Thereafter, their impacts on the crystalline domains and electrochemical performances of all the as-prepared binder-free electrodes were studied. The electrochemical analysis demonstrated that the electrode prepared at 200 °C-1h was the best compared to other electrodes. Maximum specific capacitances of 3050.95 Fg^−1^, 2489.34 Fg^−1^, 2467.70 Fg^−1^, 2252.43 Fg^−1^, and 1110.16 Fg^−1^ at 10 mVs^−1^ were achieved for the LDH-200-1h, LDH-100-2h, LDH-100-1h, LDH-100-1h30, and LDH-150-1h electrodes, respectively. Afterwards, Rct values of 0.68 Ω, 0.76 Ω, 0.78 Ω, 0.98 Ω, and 4.01 Ω were recorded for the LDH-200-1h, LDH-100-2h, LDH-100-1h, LDH-100-1h30, and LDH-150-1h electrodes, respectively. Moreover, the LDH-200-1h electrode retained 97% of its capacitance after 5000 cycles at 120 mVs^−1^. More importantly, the XRD and FTIR studies demonstrated that the excellent electrochemical performance recorded for the LDH-200-1h electrode was due to its crystalline domain which had held an important amount of water compared to its counterparts. This favored its basal spacing to be larger than those of other electrodes causing an improvement in the electrochemical performance. Furthermore, compared to the Rct values of our previously reported work, it is obvious that the Rct recorded in this study are lower, probably because of the high electrical conductivity resulting from the conductive addictive and the absence of the binder. Furthermore, < **200 °C-1h** > was noticed to be the best physical parameter to be applied for the fabrication of binder-free LDH electrodes using DMSO as a binder solvent. Overall, this confirms the reliability of the as-developed method and shows that it is a promising procedure for the industrial arena because of its simplicity and cost-effectiveness.

## Figures and Tables

**Figure 1 ijms-23-10192-f001:**
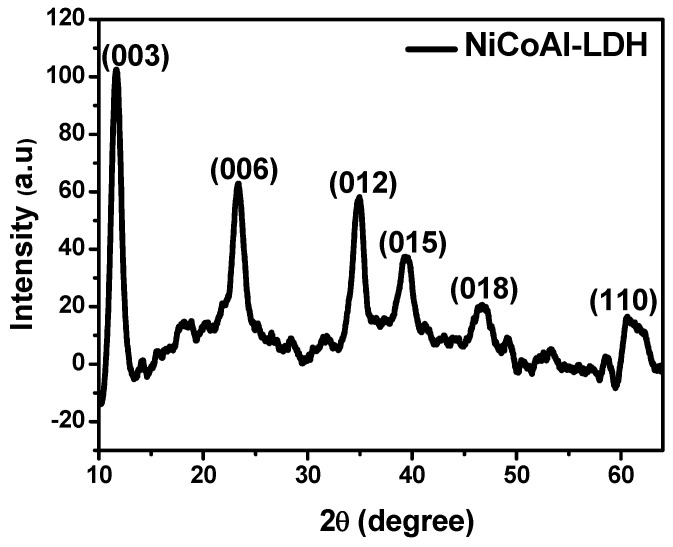
XRD patterns of the as-synthesized LDH used as electrode active material.

**Figure 2 ijms-23-10192-f002:**
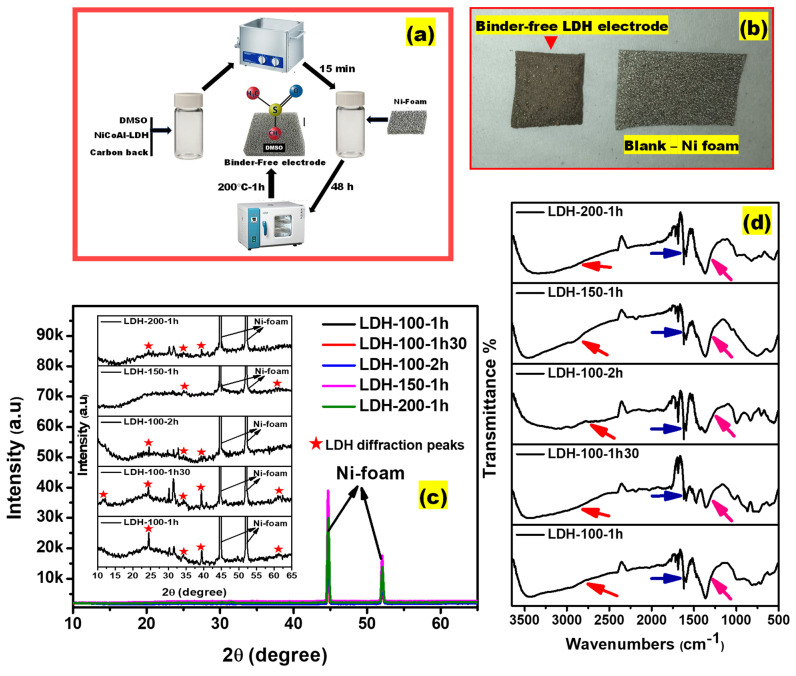
(**a**) Schematic illustration of the fabrication process of binder-free LDH electrodes; (**b**) the image of the as-fabricated binder-free LDH electrode; (**c**) XRD patterns of the LDH-100-1h, LDH-100-1h30, LDH-100-2h, LDH-150-1h, and LDH-200-1h electrodes; and (**d**) FTIR spectra of the LDH-100-1h, LDH-100-1h30, LDH-100-2h, LDH-150-1h, and LDH-200-1h electrodes.

**Figure 3 ijms-23-10192-f003:**
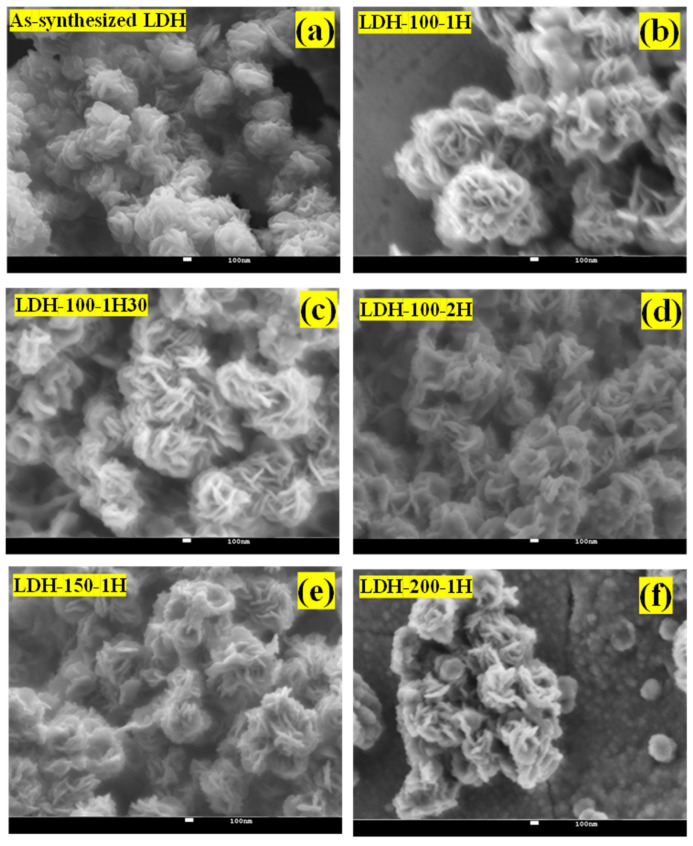
FESEM images of (**a**) the as-synthesized LDH used as electrode active material; (**b**–**f**) the LDH-100-1h, LDH-100-1h30, LDH-100-2h, LDH-150-1h, and LDH-200-1h electrodes.

**Figure 4 ijms-23-10192-f004:**
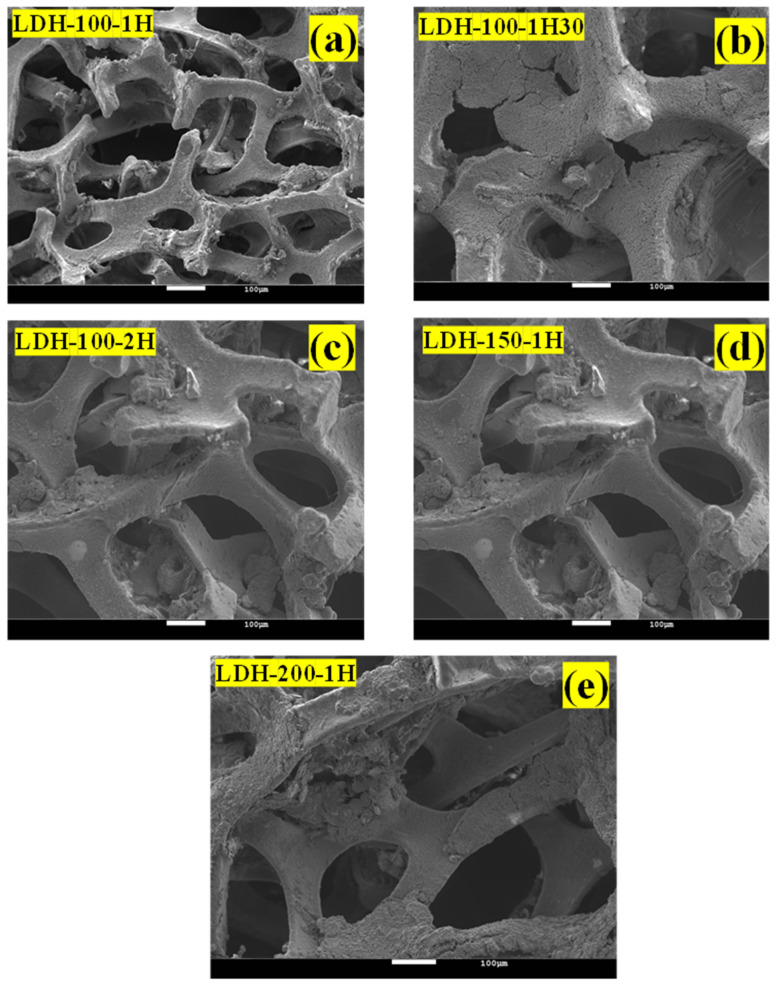
(**a**–**e**) FESEM images of the LDH-100-1h, LDH-100-1h30, LDH-100-2h, LDH-150-1h, and LDH-200-1h electrodes at a different magnification.

**Figure 5 ijms-23-10192-f005:**
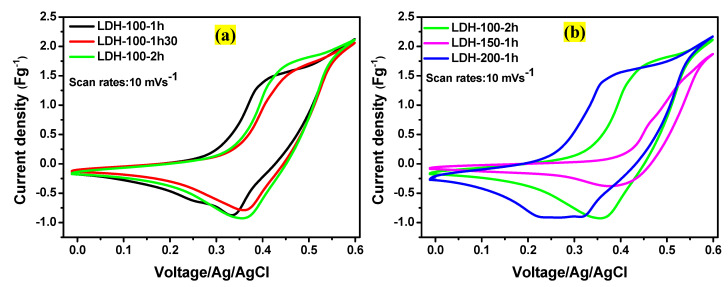
(**a**) Comparative CV curves for the LDH-100-1h, LDH-100-1h30, and LDH-100-2h electrodes at a scan rate of 10 mVs^−1^; (**b**) comparative CV curves for the LDH-100-2h, LDH-150-1h, and LDH-200-1h electrodes at a scan rate of 10 mVs^−1^.

**Figure 6 ijms-23-10192-f006:**
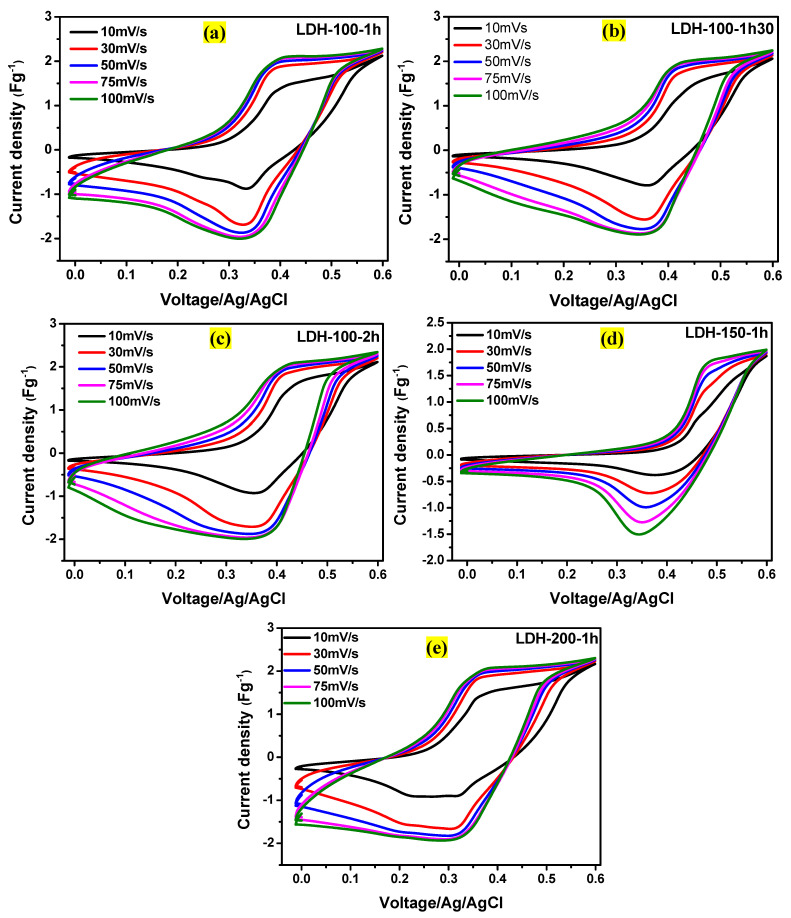
CV curves for the (**a**) LDH-100-1h, (**b**) LDH-100-1h30, (**c**) LDH-100-2h, (**d**) LDH-150-1h, and (**e**) LDH-200-1h electrodes, respectively at various scan rates.

**Figure 7 ijms-23-10192-f007:**
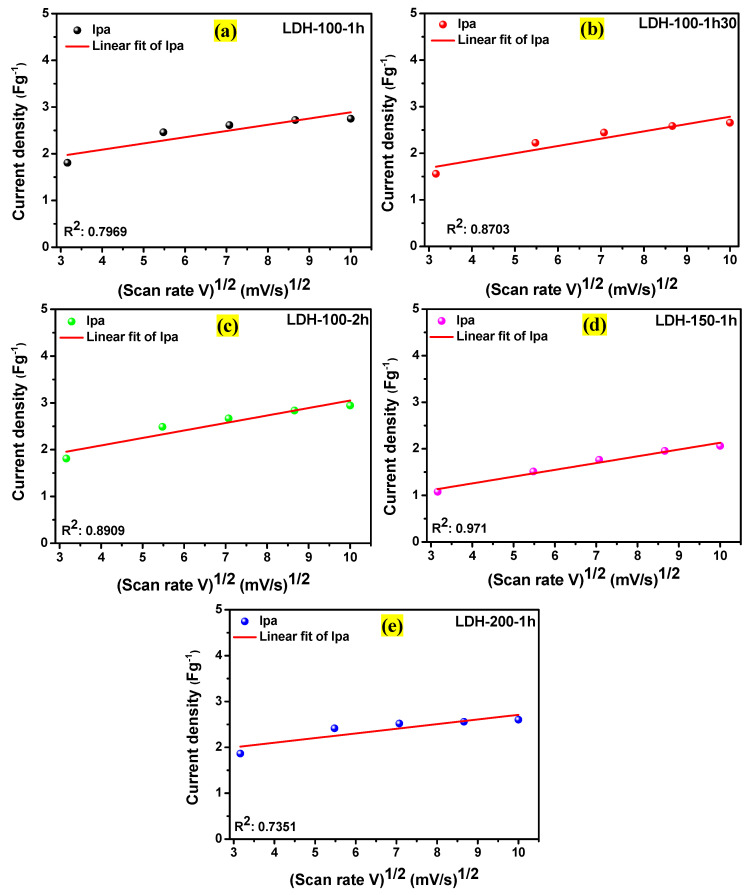
Dependence of the anodic peak current (ipa) on the square root of the scan rate for the (**a**) LDH-100-1h, (**b**) LDH-100-1h30, (**c**) LDH-100-2h, (**d**) LDH-150-1h, and (**e**) LDH-200-1h electrodes, respectively.

**Figure 8 ijms-23-10192-f008:**
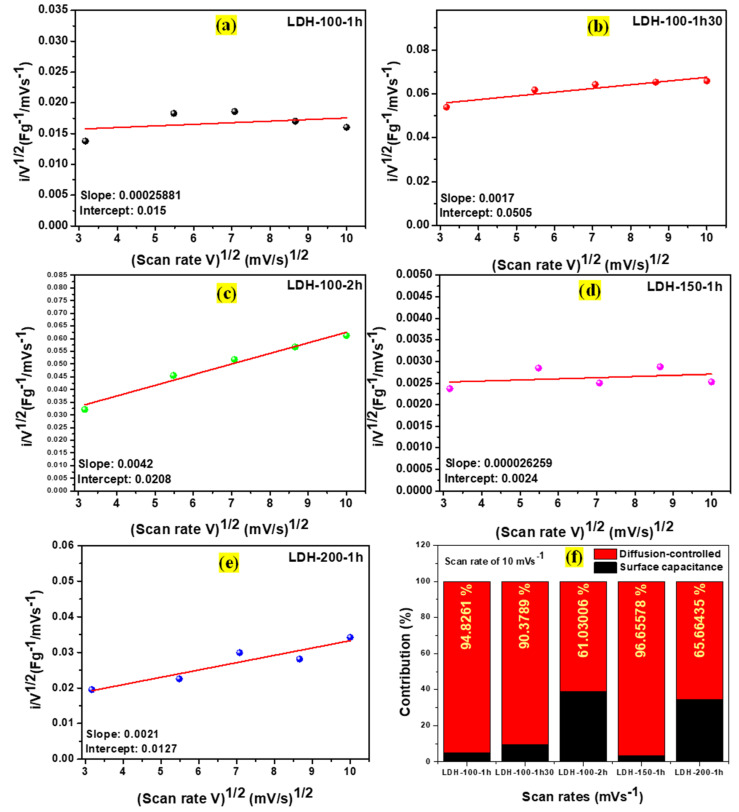
Linear relationship related to various potential levels of i/V1/2 against V1/2 for the (**a**) LDH-100-1h, (**b**) LDH-100-1h30, (**c**) LDH-100-2h, (**d**) LDH-150-1h, and (**e**) LDH-200-1h electrodes, respectively; (**f**) contribution fractions of the surface capacitance and diffusion-controlled processes for the LDH-100-1h, LDH-100-1h30, LDH-100-2h, LDH-150-1h, and LDH-200-1h electrodes, respectively, at the scan rate of 10 mVs^−1^.

**Figure 9 ijms-23-10192-f009:**
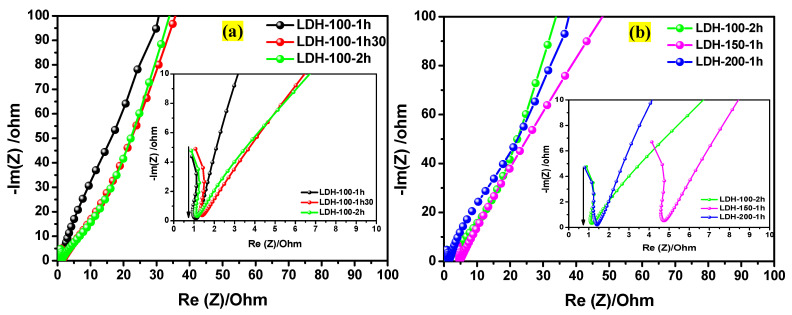
(**a**) Nyquist plots for the LDH-100-1h, LDH-100-1h30, and LDH-100-2h electrodes; and (**b**) Nyquist plots for the LDH-100-2h, LDH-150-1h, and LDH-200-1h electrodes.

**Figure 10 ijms-23-10192-f010:**
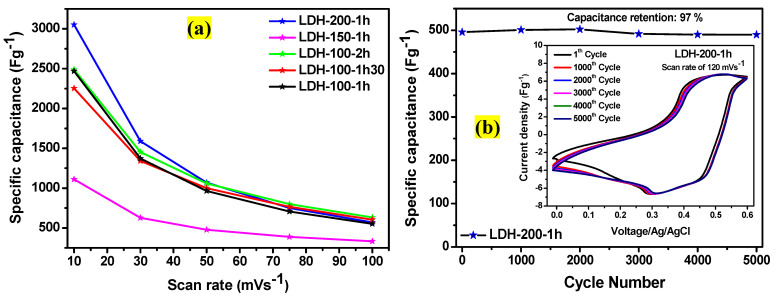
(**a**) Specific capacitances of the LDH-100-1h, LDH-100-1h30, LDH-100-2h, LDH-150-1h, and LDH-200-1h electrodes at various scan rates; (**b**) plot of capacitance retention against cycle number (insert: cyclic voltammogram of the LDH-200-1h electrode at 120 mVs^−1^).

**Table 1 ijms-23-10192-t001:** Mass-loading for all the electrodes.

Electrodes	Mass Loading (g)
LDH-100-1h	0.028
LDH-100-1h30	0.026
LDH-100-2h	0.028
LDH-150-1h	0.026
LDH-200-1h	0.029

**Table 2 ijms-23-10192-t002:** Specific capacitances calculated from the CV curves at the scan rate of 10 mVs^−1^.

Electrodes	Specific Capacitances	Scan Rate
LDH-100-1h	2467.70 Fg^−1^	10 mVs^−1^
LDH-100-1h30	2252.43 Fg^−1^	10 mVs^−1^
LDH-100-2h	2489.34 Fg^−1^	10 mVs^−1^
LDH-150-1h	1110.16 Fg^−1^	10 mVs^−1^
LDH-200-1h	3050.95 Fg^−1^	10 mVs^−1^

**Table 3 ijms-23-10192-t003:** Capacitance performance of various NiCoAL-LDH-based electrode.

Electrodes	Specific Capacitance	Electrolyte	References
NiCoS@SBA-C	1757 Fg^−1^–1 A g^−1^	6 M KOH	[55]
CuCo_2_S_4_@NiCoAl-LDH/NF	1876 Fg^−1^–1 A g^−1^	6 M KOH	[56]
Cu_2+1_O@NiCoAl-LDH	2932 Fg^−1^–0.75 A g^−1^	6 M KOH	[57]
m-LDH/NRG NHs	1877.0 Fg^−1^–1 A g^−1^	6 M KOH	[58]
NiCo_2_Al-LDH/N-GO	1136.67Fg^−1^–1 A g^−1^	2 M KOH	[51]
NiCoAl-LDH	5691.25 mF cm^−2^–1 mA cm^−2^	3 M KOH	[8]
NiCo_2_O_4_@NiCoAl-LDH	1814.24 Fg^−1^–1 A g^−1^	2 M KOH	[7]
LDH-200-1h	3050.95 Fg^−1^–10 mVs^−1^	1 M KOH	This work

**Table 4 ijms-23-10192-t004:** Rct values for all the electrodes.

	LDH-100-1h	LDH-100-1h30	LDH-100-2h	LDH-150-1h	LDH-200-1h
**Rct (Ω)**	0.78	0.98	0.76	4.01	0.68

**Table 5 ijms-23-10192-t005:** Rct values for binder-free LDH electrodes, NiCoAl-NMP and NiCoAl-DMSO electrodes.

Electrodes	Rct (Ω)
LDH-100-1h	0.78
LDH-100-1h30	0.98
LDH-100-2h	0.76
LDH-150-1h	4.01
LDH-200-1h	0.68
NiCoAl-NMP	1.5
NiCoAl-DMSO	2

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
