# Peer review of "Advantage of Dimethyl Sulfoxide in the Fabrication of Binder-Free Layered Double Hydroxides Electrodes: Impacts of Physical Parameters on the Crystalline Domain and Electrochemical Performance"

_ijms, 2022, doi:10.3390/ijms231710192_

Round 1

Reviewer 1 Report

The article presents a method to fabricate binder-free LDH electrodes using Dimethyl Sulfoxide solvent. Various physical parameters such as the temperature and time during the fabrication process on the crystalline domain and electrochemical performances of the as-obtained binder-free LDH electrodes were studied. The manuscript lacks the novelty and significance and the results are not adequately supported and discussed by important characterizations techniques. The article is not recommended for publications.

1.      Author mentioned that five pieces of nickel foam were immersed inside the vial containing the slurry and aged for 48 h at room temperature. Such tactic could not be reproducible as the mas loading on each pieces of nickel foam would be different in every experiments.

2.      In first paragraph of Introduction, author mentioned that majority of the techniques utilized in designing binder-free electrodes are complicated procedures and not simple to reproduce, which is not correct and this statement should be tuned down. Such as hydrothermal, solvothermal, chemical vapor deposition etc are not complicated methods.

3.      The manuscript lacks the physical characterizations of the LDH material such as TEM or SEM, which is very crucial to observe the morphology as well as size changes before and after electrochemical analysis.

4.      The XRD peaks varies in each sample, this should be appropriately described.

5.      Author should consider very important review article on binder-free electrode materials based on Nickel Foam. Nanoscale, 2017 9 (34), 12231-12247

Author Response

YEs

Reviewer 2 Report

Comments to authors

I write you in regards to the manuscript entitled “Advantage of dimethyl sulfoxide in the fabrication of binder-free layered double hydroxides electrodes: Impacts of physical parameters on the crystalline domain and electrochemical per- formance, submitted to the International Journal of Molecular Sciences. After going through the manuscript, I have got the belief that it is well-written in a suitable manner and the content could be interesting for the researchers in this field. Authors have provided adequate and essential data along with their interpretation. Although I am generally positive about this article, there are some small aspects where more work is requested. For the guidance of the authors, these are included below. Please consider them and deliver a minor version before reconsidering for publication.

- Line 20, line 175… analyses should be analysis

- The keywords should be more specific and related to the contents of the research.

- The novelty of the research is too vague and not clear. It should be revised to clearly reveal the gaps that the research need to be filled in.

- It is better if you could provide the SEM image of LDH

- Discussion on the XRD pattern need to be supported with the references.

- Please carefully check the English, since there still exists many grammar and errors throughout the manuscript.

- Materials used for the study must be provided in the Materials and methods section

Author Response

Yes

Round 2

Reviewer 1 Report

Author adequately addressed all the comments raised by both reviewer, I recommend the acceptance of the manuscript in present form.